# Scene Text Detection Based on Two-Branch Feature Extraction

**DOI:** 10.3390/s22166262

**Published:** 2022-08-20

**Authors:** Mayire Ibrayim, Yuan Li, Askar Hamdulla

**Affiliations:** 1School of Information Science and Engineering, Xinjiang University, Urumqi 830046, China; 2Department of Electronic Information and Automation, ABA Teachers University, Wenchuan 623002, China

**Keywords:** deep learning, residual correction branching, two-branch attentional feature fusion, two-branch feature extraction, text detection

## Abstract

Scene text detection refers to locating text regions in a scene image and marking them out with text boxes. With the rapid development of the mobile Internet and the increasing popularity of mobile terminal devices such as smartphones, the research on scene text detection technology has been highly valued and widely applied. In recent years, with the rise of deep learning represented by convolutional neural networks, research on scene text detection has made new developments. However, scene text detection is still a very challenging task due to the following two factors. Firstly, images in natural scenes often have complex backgrounds, which can easily interfere with the detection process. Secondly, the text in natural scenes is very diverse, with horizontal, skewed, straight, and curved text, all of which may be present in the same scene. As convolutional neural networks extract features, the convolutional layer with limited perceptual field cannot model the global semantic information well. Therefore, this paper further proposes a scene text detection algorithm based on dual-branch feature extraction. This paper enlarges the receptive field by means of a residual correction branch (RCB), to obtain contextual information with a larger receptive field. At the same time, in order to improve the efficiency of using the features, a two-branch attentional feature fusion (TB-AFF) module is proposed based on FPN, to combine global and local attention to pinpoint text regions, enhance the sensitivity of the network to text regions, and accurately detect the text location in natural scenes. In this paper, several sets of comparative experiments were conducted and compared with the current mainstream text detection methods, all of which achieved better results, thus verifying the effectiveness of the improved proposed method.

## 1. Introduction

Texts have become one of the indispensable means of transmitting information in the contemporary world, and there are all kinds of text information in the social scene where we live. Natural scene text detection is to locate the text area in an image through the detection network and express the text area with a polygon bounding box. The accurate detection results are beneficial to a wide range of practical applications, such as instant translation, image retrieval, scene analysis, geographic location, license plate recognition, etc., and have attracted much attention in the fields of computer vision and document analysis. In recent years, with the rapid development of the convolutional neural network (CNN), scene text detection has made great progress [1,2]. We can roughly divide the existing CNN-based text detection algorithms into two categories: regression-based methods and segmentation-based methods.

For the scene text detection algorithm based on regression, the text target is usually represented in the form of a rectangular box or quadrilateral box with a specific direction. Although the detection speed is fast, which can avoid the accumulation of multi-stage errors, most existing regression-based methods cannot accurately and effectively solve the problem of text detection due to the limited text representation (axis-aligned rectangle, rotated rectangle, or quadrilateral), especially when they are used to detect any shape text on datasets such as Total-Text [3], which is very unfavorable for subsequent text recognition in the whole optical character-recognition engine.

On the other hand, the scene text detection algorithm based on segmentation mainly locates text instances by classifying pixels. Although recent methods have made significant improvements in the task of scene text detection, and the focus of research has shifted from horizontal text to multi-directional text and more challenging arbitrary-shaped text (such as curved text), there are still challenges in the detection of arbitrary-shaped scene text due to the significant changes of its specific attributes, such as color, proportion, direction, aspect ratio, and shape, which make it obviously different from the general target objects and the different attributes of natural images, such as image-blur degree and lighting conditions.

Text in natural scenes has rich and clear semantic information. Using computer technology to extract text information quickly and accurately from scene images is one of the hot research topics in the field of computer vision and pattern recognition. Scene text detection technology is the basis for text recognition and has a wide range of applications in people’s daily lives and production. Compared with traditional OCR, text detection in natural scene images faces many difficulties and challenges, such as complex background, various text sizes and fonts, and uncertainty of image quality. In recent years, with the rapid development of deep learning technology, the method of deep learning has achieved remarkable results in the task of text detection, and the existing convolutional neural network already has a good representation ability. However, the network’s insufficient receptive field, weak positioning ability, and inaccurate positioning of texts will lead to false detection or missed detection when detecting long or large texts. On the other hand, although the feature pyramid network can fuse features of different scales, the high-level semantic information of small-scale texts has been lost at the top of the network, which leads to the weak detection ability of the model for multi-scale texts.

Text information in natural scenes usually has the characteristics of diversity and irregularity and the complexity of arbitrary-shaped text detection in natural scenes. Due to the manual feature design, the traditional text detection methods of natural scenes lack robustness, while the existing text detection methods based on deep learning have the problem of losing important feature information in the process of extracting features from various layers of networks. The segmentation-based text detection method has been one of the most popular detection methods recently, and the segmentation results can more intuitively describe scene texts of various shapes. The original DB (differentiable binarization) algorithm [4] simplifies the post-processing process by using the differentiable binarization algorithm, which solves the gradient non-differentiable problem caused by training and improves the efficiency of scene text detection. However, inadequate use of semantic and spatial information in the network limits the network’s classification and localization capabilities. Despite the advantages of segmentation-based algorithms in detecting arbitrary-shaped text, the lack of sufficient contextual information can also result in false positives or missed detections.

In order to make the DBNet text detection network acquire deeper semantic information and clear key text features in the process of feature extraction, this paper improves the feature extraction network based on the DBNet algorithm and introduces residual correction branch (RCB) and two-branch attention feature fusion (TB-AFF) modules to form a new text detection network. Its core framework is shown in Figure 1. In this paper, a text detection model based on the lightweight network ResNet is proposed. The improved ResNet lightweight feature extraction network and better feature fusion method are used to effectively fuse features of different depths to guide segmentation. In this model, the residual correction branch (RCB) is used to expand the receptive field, to improve the ability of obtaining context information, and, thus, to obtain the context information of a larger receptive field. At the same time, in order to improve the efficiency of using the features of the model, a two-branch attention feature fusion (TB-AFF) module is added to the FPN structure. By combining global and local attention mechanisms, the text area can be accurately located and the text position in the natural scene can be accurately detected. Finally, through the differentiable binarization module, the binarization process is added to the training process of the model, and the binarization threshold is set adaptively, so that the probability graph generated by the segmentation method is converted into a text region, and a better text detection effect is achieved. The whole model not only guarantees the quality of feature extraction, but also achieves a good balance between speed and accuracy because it belongs to a lightweight network. The proposed method is tested on the dataset, and, without sacrificing the speed, it expands the receptive field of the network, learns more detailed text location information, and further locates the text area accurately. Experiments show that the improved method is obviously superior to previous methods in accuracy. On ICDAR2015 [5], Total-Text, and MSRA-TD500 datasets [6], the detection method proposed in this paper achieves comparable results in speed and accuracy and improves the multi-scale detection performance of the model.

To demonstrate the effectiveness of our proposed framework, we conducted extensive experiments on three competitive benchmark datasets, including ICDAR 2015, Total-Text, and MSRA-TD500. In these datasets, Total-Text is specifically designed for curved text detection. Therefore, the experimental results on the MSRA-TD500 and Total-Text datasets show that this method has high flexibility in complex situations (such as multi-language, curved text, and arbitrary-shaped text). Specifically, on the Total-Text dataset with arbitrary-shaped text, we surpass the results of most of the most advanced methods with an absolute advantage, and our model achieves considerable performance. In addition, the framework proposed in this paper has achieved good performance on the multidirectional text dataset ICDAR 2015.

In this paper, the residual correction branch (RCB) and two-branch attention feature fusion (TB-AFF) modules are used for lightweight feature extraction. In order to make up for the deficiency of the feature-extraction ability of a lightweight network, the residual correction branch (RCB) is embedded into the backbone network to enhance its feature-extraction ability. In addition, this paper also proposes a two-branch attention feature fusion (TB-AFF) module, which is used to enhance the feature expression of multi-scale scene texts and improve the accuracy of its detection.

To sum up, the main contributions of this paper are as follows: (1) A residual correction branch (RCB) is proposed. In order to make up for the shortcomings of a lightweight network in feature-extraction ability and receptive field, the residual correction branch (RCB) is embedded in the backbone network to enhance its feature-extraction ability. This is a more efficient method than previous methods, has a generic structure, and is less computationally intensive. (2) The proposed two-branch attention feature fusion (TB-AFF) module is used to enhance the feature expression of multi-scale scene texts, so as to improve the accuracy of its detection. It can effectively solve the detection problem of scene texts with arbitrary shapes and improve the performance of scene text detection. (3) We achieve state-of-the-art performance on several benchmarks including different forms of text instances (directed, long, multilingual, and curved), demonstrating the superiority of our newly designed module.

## 2. Related Work

In recent years, with the rise of deep learning, various scene text detection algorithms based on neural networks have been proposed, one after another. Using convolutional neural network to automatically learn text features frees researchers of scene text detection from the tedious process of manually designing features, making remarkable progress in scene text detection technology. At present, scene text detection methods based on deep learning can be divided into two categories: regression-based methods and segmentation-based methods.

The algorithm of scene text detection based on regression is mainly inspired by Faster R-CNN, SSD [7], Mask R-CNN [8], FCIS [9], etc. Usually, some text boxes need to be set in advance, and the convolutional neural network is used to determine whether they overlap with the text area and adjusts their size and position to accurately locate the text. Based on Faster R-CNN, Zhong et al. put forward the DeepText algorithm [10]. By introducing the Inception structure into RPN network, the candidate text boxes with overlapping areas are deleted by voting instead of the non-maximum suppression method, and the final detection result is obtained. Tian et al. proposed an algorithm based on Faster R-CNN, which has strong detection function for fuzzy text and multilingual text [11]. Shi et al. proposed a simple and effective multi-angle text detection algorithm SegLink based on SSD [12]. Different from ordinary objects, texts are usually displayed in irregular shapes with different aspect ratios. To solve this problem, Liao et al. proposed an end-to-end natural scene text detection method, TextBoxes, in 2017 [13]. Liao et al. modified the convolution kernel and text box based on TextBoxes and put forward TextBoxes++ [14]. The network architecture of TextBoxes++ is like that of TextBoxes, replacing the global average pooling in the last layer with a convolutional layer and introducing angle prediction to achieve detection of text in any direction. Ma et al. proposed a rotated region proposal network (RRPN) through Faster-RCNN detection algorithm to solve the problem that RPN can only detect horizontal text [15]. Based on Faster R-CNN, Jiang et al. improved the RoI pool layer and prediction header and proposed the Rotational Region CNN (R2CNN) [16]. From an instance-aware perspective, Dai et al. proposed an FTSN based on the instance segmentation framework FCIS [9]. Most of the existing regression-based methods rely on preset text boxes. However, in a natural scene, the text direction is changeable, and the size and aspect ratio change dramatically. In order to make the preset text box overlap with the text area, many methods add text boxes with various directions, sizes, and aspect ratios, but this undoubtedly increases the complexity and calculation of the method. Segmentation-based approaches draw mainly on semantic segmentation methods, which treat all pixels in a text bounding box as positive sample regions, describe text regions by employing different representations, and then reconstruct text instances through specific post-processing. The greatest benefit of these methods is the ability to extract text of an arbitrary shape. Zhou et al. proposed a new text detection algorithm, East, based on FCN, in 2017 [17]. Deng et al., inspired by SegLink, proposed to detect scene text by example segmentation [18]. To solve the problem that irregular–shaped text is difficult to detect, Long et al. proposed a TextSnake algorithm that can detect arbitrary-shaped scene text through the text centerline [19]. In [20], Wang et al. proposed the Progressive Scale Expansion Network (PSENet) text detection method, which takes a breadth-first traversal-based approach to progressively expand text kernels to reconstruct text instances. Wang et al. put forward the Pixel Aggregation Network (PAN), based on [21]. He et al. proposed to use the Direct Regression Network (DDR) to detect multi-directional scene text [22]. Tian et al. [23] regards each instance text as a cluster and clusters pixels by embedding mapping. TextField uses depth domain to link adjacent pixels and generate candidate text parts [24]. Based on pyramidal instance segmentation, PMTD discards the pixel-by-pixel binary prediction segmentation method and combines shape and location information to mitigate boundary discontinuities caused by inaccurate labeling [25]. The classical text detection algorithm is limited by convolution kernel receptive field, so it cannot detect long text well. To address this problem, Baek et al. [26] proposed the Character Region Awareness for Text Detection (CRAFT) algorithm based on character-probability prediction. Huang [27] proposed the Mask-Pan algorithm, based on the Mask Rcnn algorithm and the pyramid attention network. The pyramid attention network enables the model to better focus on the context information of the image text for location and classification. Xie [28] and others proposed the SPCNET network, based on Mask-Rcnn, to add an attention mechanism. This method introduced the text-context module and attention mask, which made the algorithm better integrate the intermediate features of semantic segmentation with the detection features and improved the detection accuracy. DBNet adopts adaptive binarization for each pixel, derives the binarization threshold from network learning, and adds the binarization steps to the network for training. A differentiable binarization method is proposed to solve the nondifferentiable problem, to reduce the post-processing steps. The network runs fast and is sensitive to background interference. Based on DBNet, our method makes up for the shortcomings of the feature-extraction ability and receptive field of a lightweight network and greatly improves the performance of text detection.

## 3. Method

### 3.1. Residual Correction Branch (RCB)

In this paper, we do not design a complex network architecture to enhance the representation of text features but to improve the backbone network without adjusting the model architecture, to achieve the purpose of improving the performance of the whole network. Therefore, this paper proposes a novel residual correction branch (RCB) as an effective method, as shown in Figure 2, to help convolutional networks learn the discriminant representation of text feature information.

This module mainly enhances the detection capability of the whole network by increasing the effective receptive field of the network. Specifically, firstly, the input X passes through a conventional convolution layer, BN layer, and Relu activation function, and then it is sent to two different branches, $x_{1}$ and $x_{2}$, to obtain the feature information in different spaces. On the first branch, $x_{1}$, a simple convolution and BN operation are used to extract features, the purpose of which is to retain the spatial information on the main branch of the original backbone network, that is, the input feature $x_{1}$ is convolved to get the output feature $y_{1}$. In the other branch, $x_{2}$, firstly, the input is downsampled by averaging pool, and the receptive field of CNN is increased by downsampling. Then, the obtained feature graph is subjected to convolution and BN operation, and then the nearest neighbor interpolation up-sampling is performed to restore the input size. Finally, the output feature, $y_{2}$, is obtained by Sigmoid activation function. $x_{1}$ and $x_{2}$ branches are used to extract enough text information features in parallel. Next, the output features, $y_{1}$ and $y_{2}$, of these two branches are multiplied element by element and then added with the original input X, and the final output feature Y of the network is obtained through the Relu activation function. Thanks to the design of this double-branch structure, the receptive field of each spatial location is effectively expanded, so that every pixel in the space has the information of its surrounding area, thus paying attention to more contextual information. The specific process is as follows:

Given the original input X, the branches $x_{1}$ and $x_{2}$ are obtained as follows:(1)$$x_{1},x_{2}=Relu\left(BN\left(Conv\left(X\right)\right)\right)$$
acquisition of output feature $y_{1}$:(2)$$y_{1}=\mathrm{BN}\left(\mathrm{Conv}\left(x_{1}\right)\right)$$

Given the input branch $x_{2}$, the average pooling with a filter size of 4 × 4 and a step size of 4 are adopted, as follows:(3)$$x_{2}^{\prime }={\mathrm{AvgPool}}_{r}\left(x_{2}\right)$$

${\mathrm{AvgPool}}_{r}$ represents the average pool function with downsampling r times. Through the average pool operation, the image scale is reduced and the network receptive field is expanded. This feature can help CNN to generate more discriminating feature expressions and extract richer information.

Acquisition of output feature $y_{2}$:(4)$$y_{2}=\mathrm{Sigmoid}\left(\mathrm{NN}\left(\mathrm{BN}\left(\mathrm{Conv}\left(x_{2}^{\prime }\right)\right)\right)\right)x_{2}^{\prime }={\mathrm{AvgPool}}_{r}\left(x_{2}\right)$$
where NN(·) is the nearest neighbor interpolation upsampling, and Sigmoid activation function can increase the nonlinearity of neural network model, so as to increase the fitting ability of nonlinear relationship of samples.

Acquisition of output feature Y:(5)$$Y=Relu\left(X⊕\left(y_{1}⊙y_{2}\right)\right)$$

The module has two parallel branches, the two branches are carried out independently, and the outputs of each branch are combined as the output of the final network. Therefore, the whole network can generate a larger receptive field, fully obtain the context information of the text features in the image, and help to capture the whole text area well. On the other hand, the residual correction branch does not need to collect the whole global context information, but only considers the context information around each spatial position, which avoids some pollution information from irrelevant areas (non-text areas) to some extent and can accurately locate the text areas. Moreover, as can be seen from Figure 2, RCB does not need to rely on any additional learnable parameters, has strong versatility, can be easily embedded into modern classification networks, is suitable for various tasks, and is convenient to use.

### 3.2. Two-Branch Attention Feature Fusion (TB-AFF) Module

As we all know, attention plays a vital role in computer vision. As the text has its own texture features, it is worth considering what kind of attention module should be designed to perfectly match the text features. The attention mechanism in deep learning originated from the human visual attention mechanism [29,30]. For example, SENet compresses global spatial information into channel descriptors to capture channel correlation [31]. The way to calculate attention is to average the pixel values of each channel, and then, after a series of operations, normalize them with sigmoid function. SENet is specifically expressed as follows.

Given the intermediate feature X∈ℝC×H×W of C channel and the feature graph of size H×W, the channel attention weight w∈ℝC in SENet can be calculated as:(6)w=σ(g(X))=σ(ℬ(W2δ(ℬ(W1(g(X))))))

In which g(X)∈ℝC represents the global context feature, and g(X)=1H×W∑i=1H∑j=1WX[:,i,j] is the global average pooling (GAP). δ represents rectification linear unit (Relu), B represents batch standardization (BN), and σ is Sigmoid function. This is achieved by having two fully connected (FC) layers. W1∈ℝCr×C is a dimension-decreasing layer, W2∈ℝC×Cr is a dimension-increasing layer, and r is the channel-reduction rate. Here, the attention of this channel squeezes each feature graph of size H×W into a scalar. This extremely rough description tends to emphasize the global distribution of large targets, and it is effective for large-scale targets. However, the effect is not very good for small-scale targets, so small targets are often ignored. However, compared with the whole image, the proportion of text is small, which belongs to our small target detection. Therefore, global channel attention may not be the best choice. Here, this paper proposes a multi-scale attention fusion network (MSAFN), which uses attention to fuse text features.

For the feature pyramid network FPN, the deeper the features, the more channels there are. However, the feature maps of each layer propagate from top to bottom during fusion, so the deeper the feature maps, the more channels will be reduced. The reduction in channels will inevitably lead to the loss of feature information, and the higher-level features will often lose more feature information. Therefore, in order to retain more information of text features, here we propose a unified and universal scheme based on the structure of FPN, that is, the Double Branch Attention Feature Fusion (TB-AFF) module, which is a multi-scale feature extraction module. This paper mainly discusses the way of feature fusion and better fuses features of different semantics or scales. As far as we know, the two-branch attention feature fusion (TB-AFF) module has never been discussed before. By aggregating multi-scale text feature information along the channel dimension, combining local attention information and global attention information, this module emphasizes the larger target with more global distribution and highlights the smaller target with more local distribution, thus alleviating various problems caused by the change of text scale, improving the representation ability of text feature information, and further improving the performance of text detection. The scheme of our proposed architecture is shown in Figure 3.

In this part, we will describe the proposed two-branch attention feature fusion (TB-AFF) module in detail. The key idea of B-AFF is to extract global and local attention weights by changing the dimension size and using two branches with different scales, to achieve local and global attention on multiple scales. TB-AFF module structure is relatively simple. Global feature branch extracts global feature attention by using global average pooling and pointwise conv (that is, ordinary convolution with convolution kernel 1). The local feature branch also uses pointwise conv to extract local feature attention, in order to preserve details. For SENet, only the global channel attention is used, while the proposed TB-AFF module also aggregates the local feature attention, which helps the whole network to contain less background clutter and irrelevant areas and is more conducive to the detection of small targets.

In this paper, the characteristic layers $C_{4}$ and $C_{5}$ are taken as examples. We choose to perform the initial feature fusion for the two input features, $C_{4}$ and $C_{5}$, that is, perform the original operation process of FPN. Given two inputs, $C_{4}$ and $C_{5}$, the high-order feature $C_{5}$ is upsampled by linear interpolation, and then it is fused with the next-order feature $C_{4}$, so that feature information from different levels of semantics is fused in each layer of feature map. In the above process, the initial feature fusion is just the addition of simple corresponding elements, and then we will carry out more detailed operations.

Given the feature *X*, after the initial fusion of input C4 and C5, the one-dimensional attention g(x)∈ℝC×1×1 obtained on the global feature branch is extracted by point-by-point convolution:(7)g(X)=PWConv(PWConv(Avg(X)))

Avg stands for global average pooling, and PWConv stands for point-by-point coupon product. Global attention information can be calculated by global feature branch, because global average pooling is used here, which makes the obtained feature information contain global information. Point-by-point convolution is also used here, and the convolution direction is changed by gradually compressing the channel, to assign more weight to the text area with high response. Similarly, the three-dimensional attention L(X)∈ℝC×H×W obtained on the local feature branch is also extracted by point-by-point convolution, and the formula is as follows:(8)L(X)=PWConv(PWConv(X))

PWConv stands for point-by-point convolution. L(X) has the same shape as the input feature X and can retain and highlight details in low-level features.

The global one-dimensional attention g(X) and the local three-dimensional attention L(X) are collected, the feature X′∈ℝC×H×W that needs attention is defined, and the formula is as follows:(9)X′=L(X)⊕g(X)
where ⊕ represents addition and represents broadcast addition operation. The main reason is that the global one-dimensional attention uses the global average pooling operation, and the obtained feature height-width shape is 1 × 1, while the local three-dimensional attention keeps the same height-width dimension *H* × *W* as the feature X, so the broadcast operation is needed when the two are added together.

The obtained feature $X^{\prime }$ is activated by sigmoid function, and, finally, the dot multiplication operation is performed with the smaller feature layer (here, $C_{4}$) in the original input. The formula is as follows:(10)$$P_{4}=C_{4}⊙\left(sigmoid\left(X^{\prime }\right)\right)$$

Here, using sigmoid activation function can make the value of each element between [0, 1], which can enhance useful information and suppress useless information.

The original input of the network is 640 × 640 × 3, and, after the output of the residual correction branch (RCB), five feature maps (taking ResNet18 as an example) are obtained, namely Y_1_, Y_2_, Y_3_, Y_4_, and Y_5_. Due to the large size of the feature map Y_1_, the parameters will increase dramatically, which leads to the high complexity of the network. Therefore, we abandon Y_1_ and choose the last four feature maps Y_2_–Y_5_, the sizes of which are shown in Table 1. In order to reduce the parameter quantity and complexity of the network model, we reduce the dimension of the feature maps Y_2_–Y_5_ and change the number of channels to 64, so as to obtain the input of the two-branch attention feature fusion (TB-AFF) module, namely C_2_–C_5_. The purpose of TB-AFF proposed in this paper is to introduce richer details and global information for high-level features and low-level features, so that the extracted features can better highlight the local and global feature of text examples, thus improving the information representation ability. Therefore, we keep the input and output dimensions of the two-branch attention feature fusion (TB-AFF) module consistent, that is, P_2_ = C_2_, P_3_ = C_3_, P_4_ = C_4_, and P_5_ = C_5_.

To sum up, the two-branch attention feature fusion (TB-AFF) module combines local and global feature information and uses feature maps with different scales to extract attention weights and adjust the text position. The main contributions are as follows:(1)TB-AFF module focuses on the size of attention by point-by-point convolution, instead of convolution kernels with different sizes. Point-by-point convolution is also used to make TB-AFF as lightweight as possible.(2)TB-AFF module is not in the backbone network but is based on feature pyramid network FPN. It aggregates global and local feature information, strengthens contact with contextual feature information, and updates the text area.

### 3.3. Differentiable Binary Module

According to the probability graph $P\in R^{H\times W}$ generated by the segmentation network, h and w represent the height and width of the input image, respectively. To transform the probability graph into a binary graph, the binarization function is essential. The standard binarization function is shown in Formula (11), and the pixel with a value of 1 is considered as an effective text area.
(11)$$B_{i,j}=\{\begin{array}{l} 1,P_{i,j}⩾t\\ 0,\;\mathrm{other}\; \end{array}$$
where t is the set threshold, and (i,j) represents the coordinate point. Equation (11) is a standard binary function, which is non-differentiable, so it cannot be optimized by dividing the network. In order to solve the problem that the binarization function is not differentiable, this paper uses Formula (12) to make differentiable binarization:(12)$$B_{i,j}^{\prime }=\frac{1}{1+e^{-k\left(P_{i,j}-T_{i,j}\right)}}$$
where $B^{\prime }$ is an approximate binary graph, T is an adaptive threshold graph learned from the network, and k is an amplification coefficient. In the training process, the role of k is to increase the propagation gradient in the back propagation, which is friendly to the improvement of most prediction error areas and conducive to the generation of more significant predictions. In this paper, set k = 50.

Specifically, the probability map (P) and the threshold map (T) are predicted by using features, and the approximate binary map is obtained by combining the probability map and the threshold map according to the differentiable binary module, so that the threshold of each position of the image can be predicted adaptively. Finally, the text detection box is obtained from the approximate binary image. The structure of microbinarization is shown in Figure 4. The green path represents the standard binarization process, and the red path is the differentiable binarization used in this paper.

### 3.4. Loss Function

Loss function plays a vital role in deep neural network. Here, we use $L_{1}$ loss function and binary cross entropy loss function to optimize our network. The loss function of this paper consists of three parts in the training process: segmentation map loss $L_{s}$, binarization on map $L_{b}$ and threshold map $L_{t}$:(13)$$L=L_{S}+\alpha \times L_{b}+\beta \times L_{t}$$
where α and β are weight parameters, α is set to 1, and β is set to 10. Among them, binary cross entropy loss function is used for probability map loss $L_{s}$ and binary map loss $L_{b}$, and its formula is as follows. Hard negative mining is also used to overcome the imbalance of positive and negative samples.
(14)$$L_{s}=L_{b}=\sum_{i\in S_{l}}y_{i}\log x_{i}+\left(1-y_{i}\right)\log\left(1-x_{i}\right)$$

Among them, $S_{l}$ represents the sampling sample of the image with the ratio of positive and negative samples of 1:3. For the loss $L_{t}$ of the adaptive threshold map, the $L_{1}$ loss function is adopted, and its formula is:(15)$$L_{t}=\sum_{i\in R_{d}}\left|y_{i}^{*}-x_{i}^{*}\right|$$
where $R_{d}$ is the index of pixels in this area, and $y^{*}$ is the label of adaptive threshold map.

## 4. Experimental Results and Analysis

In this paper, three challenging public datasets are tested, namely, the multidirectional text dataset ICDAR2015, curved text dataset Total-Text, and multilingual text dataset MSRA-TD500. The visualization results of this method on different types of text examples are shown in Figure 5, including curved texts (e) and (f), multi-directional texts (a) and (b), and multilingual texts (c) and (d). For each cell in Figure 5, the second, third, and fourth columns are probability graph, threshold graph, and binarization graph, respectively.

The ICDAR 2015 dataset contains 1000 training images and 500 test images. All the images are taken automatically by the camera, and the shooting angle is not adjusted, so it is very random, and there is tilt and blur. Therefore, the text may appear in any direction and any position. At the same time, the text appears randomly in a certain position in the image, and the dataset has not been adjusted to improve the image quality, in order to increase the difficulty of detection.

The Total-Text dataset is a public dataset used to detect bent texts. It contains bent texts of commercial signs in real scenes, and the language to be detected is English. There are 1555 pictures, 1255 training images, and 300 test images.

The MSRA-TD500 dataset belongs to a multi-language and multi-category dataset, including Chinese and English, and contains 500 pictures, 300 training images, and 200 test images. These images are mainly taken indoors (offices and shopping malls) and outdoors (streets) with cameras. Indoor images include signs, house numbers, and warning signs. Outdoor images include guide boards and billboards with complex backgrounds.

### 4.1. Training Configuration

In this paper, Python 3.7 is used as the programming language, and Pytorch1.5 is used as the deep learning framework. All the experiments were carried out on TITAN RTX. The initial learning rate is set to 0.007. The training process includes two steps: first, we use SynthText dataset [32] to train the network for 100,000 iterations, and then we fine-tune the model on the benchmark real dataset 1200 times. We only use the official training images of each dataset to train our model, with a weight attenuation of 10-4 and a momentum of 0.9. The optimizer used for training is Adam [33]. The training batch size is set to 16. The training data are enhanced by randomly rotating the angle, cropping, and flipping in the range of (−10, 10), and all the pictures are readjusted to 640 × 640.

It is worth noting that the fuzzy text marked “ignored” is ignored in the training process. In the preprocessing stage of the network, the labels of probability graph and threshold graph are created based on the labels of training datasets. Since small text areas are not easy to detect, some too-small text areas will be ignored in the process of creating labels (for example, the minimum side length of the smallest rectangle of the text area is less than 3 or the polygon area is less than 1). Therefore, during the training process, a small part of the text marked “ignored” will be discarded.

Since the test images of different scales have great influence on the detection effect, the aspect ratio of the test images is kept in the reasoning stage, and the size of the input images is adjusted by setting an appropriate height for each dataset.

### 4.2. Experiment and Discussion

In order to better prove the realization of each module proposed in this paper, we have carried out detailed ablation experiments on the multi-directional text dataset ICDAR2015, curved text dataset Total-Text, and multilingual text dataset MSRA-TD500. Three main performance parameters, precision (P), recall (R), and comprehensive evaluation index (F), are considered to evaluate the detection performance of the model, which proves the effectiveness of the residual correction branch (RCB) and the two-branch attention feature fusion (TB-AFF) modules proposed by us. During the training process, the experiment was conducted in the same environment, and the place marked “√” indicated that the method was used. The results are listed in Table 2, Table 3 and Table 4.

As can be seen from Table 2, on the ICDAR2015 dataset, after adding the RCB module, the recall rate and F value exceed the original DB model results by about 4.68% and 1.56%, respectively. After adding the TB-AFF module, the recall rate and F value exceed the original DB model results by about 4.82% and 2.03%, respectively. At the same time, by adding these two modules, the method achieves 79.48% recall rate, 87.26% accuracy rate, and 83.19% F value in natural scene text image detection, which ensures the integrity of text information in the process of text detection. Compared with the results of the original model, the recall rate and F value increased by 5.68% and 2.39%, respectively, under the same accuracy. Therefore, the detection performance of the network combining these two modules is better than that of the network using the RCB module or TB-AFF module alone.

As can be seen from Table 3, on the Total-Text dataset, compared with the local original DB model reproduction results, the introduction of the RCB module improves the recall rate and F value by about 4.56% and 2.12%, respectively. After the introduction of the TB-AFF module, the recall rate and F value increased by about 5.33% and 2.10%, respectively. At the same time, by introducing the two modules, this method achieves 78.95% recall rate, 87.37% accuracy rate, and 82.95% F value in natural scene text image detection. Compared with the results of the original model, the recall rate and F value are improved by 5.15% and 2.15%, respectively, under the same accuracy rate. Therefore, the detection performance of the network combining these two modules is better than that of the network using the RCB module or TB-AFF module alone.

As can be seen from Table 4, on the MSRA-TD500 dataset, compared with the local original DB model reproduction results, after the introduction of the RCB module, the recall rate and F value are increased by about 7.82% and 3.35%, respectively. After the introduction of the TB-AFF module, the recall rate and F value increased by about 6.78% and 2.95%, respectively. At the same time, by introducing the two modules, the method achieves 83.33% recall rate, 88.02% accuracy rate, and 85.61% F value in natural scene text image detection. Compared with the results of the original model, the recall rate and F value are increased by 9.53% and 4.81%, respectively, under the same accuracy. Therefore, the detection performance of the network combining these two modules is better than that of the network using the RCB module or TB-AFF module alone.

It can be seen from the above observation that in the residual correction branch (RCB) module, we introduce the average pool down-sampling operation to establish the connection between positions in the whole pool window. The experimental results show that using the 18-layer backbone network and the proposed RCB can greatly improve the baseline performance, and the results are obviously improved. This phenomenon shows that the network with a residual correction branch can generate richer and more distinctive feature representations than the original ordinary convolution, which is helpful to find more complete target objects and can be better confined to semantic areas, even though their sizes are small. At the same time, in order to overcome the semantic and scale inconsistency between input features, our two-branch attention feature fusion (TB-AFF) module combines local feature information with global feature information, which can capture context information better. The experimental results show that the multi-scale attention fusion network (MSAFN) with the dual-branch attention feature fusion (TB-AFF) module can improve the performance of advanced networks with a small parameter budget, which indicates that people should pay attention to feature fusion in deep neural networks, and a proper attention mechanism of feature fusion may produce better results. It is further explained that instead of blindly increasing the depth of the network, it is better to pay more attention to the quality of feature fusion.

We have carried out relevant experiments on the downsampling factor r, and the experimental results are shown in Table 5. We find that the smaller the value of r, the greater the complexity of Flops is. When r = 3,4,5, the complexity of its network Flops is similar. When r = 4, its f value reaches the relatively optimal result. In addition, we also verified it on Ours-Resnet-50 and found that when r = 4, the Flops and F values were well-balanced. Among them, the experimental results in Table 5 were carried out on the improved Resnet18, and Flops was calculated by inputting 3 × 640 × 640.

In natural scene text detection, most cases appear as characters or text lines. The residual correction branch (RCB) designed by us increases the receptive field of the network by downsampling the feature map, thus modeling the context information around each spatial location and making the network detect the text information in the image more accurately and completely. The ablation experiment results also verify this point, which shows that the RCB proposed by us is effective.

Figure 6 shows the visualization results of baseline and the method in this paper. For each unit in the graph, the second column is the probability graph, the third column is the threshold graph, and the fourth column is the binary graph. From the experimental results, the residual correction branch (RCB) and the double branch attention feature fusion (TB-AFF) modules play an important role in text feature extraction in model training, effectively enhancing the model’s attention to text features, making effective use of the extracted text features and improving the detection accuracy of scene text to some extent.

We compare the proposed method with other advanced methods on different datasets, including the multi-directional text dataset ICDAR2015, curved text dataset Total-Text, and multilingual text dataset MSRA-TD500. The experimental results are shown in Table 6, Table 7 and Table 8.

The algorithm in this paper is compared with other algorithms on the Total-Text curved text dataset, and the results are shown in Table 6. Our model outperforms segmentation-based algorithms such as the TextSnake algorithm, PSENet algorithm, and TextField algorithm in three evaluation indexes. Ours-ResNet-18 (800 × 800) achieved 78.95% recall, 87.37% accuracy, and 82.95% F value, which surpassed the original model DB-resnet-18 (800 × 800) by about 0.67%, 3.55%, and 2, respectively. Ours-ResNet-50 (800 × 800) achieved 82.19% recall rate, 88.06% accuracy rate, and 85.03% F value, which were about 3.76% and 3.79% higher than the original model DB-ResNet-50 (800 × 800), respectively. The above experimental results show that this model can adapt to any shape of curved text detection, and, in most cases, the method proposed in this paper is obviously superior to other methods.

We also tested the parameters and complexity of other models in Table 6, and the test results are shown in Table 6. As can be seen from Table 6, Ours-ResNet-18 (800 × 800) has increased a little parameter quantity and complexity compared with the baseline and achieved a performance improvement of 2.25%. For Ours-ResNet-50 (800 × 800), although the parameters and complexity of the model have increased, the performance has improved by about 4%. Compared with PSE-1s, our model only needs fewer parameters and complexity to achieve better performance.

On the multidirectional text dataset ICDAR2015, the comparison results between our algorithm and other algorithms are shown in Table 7. Ours-ResNet-18 (1280 × 736) achieved 79.48% recall rate, 87.26% accuracy rate, and 83.19% F value, which exceeded the original model DB-ResNet-18 (1280 × 736) by about 5.68% and 2.39%, respectively. Ours-ResNet-50 (1280 × 736) achieved 79.83% recall, 87.82% accuracy, and 83.63% F value, which surpassed the original model DB-ResNet-50 (1280 × 736) by about 2.03% and 0.73%, respectively. Ours-ResNet-50 (2048 × 1152) achieved a recall rate of 84.26%, an accuracy rate of 88.21%, and an F value of 86.19%, which surpassed the original model DB-resnet-50 (2048 × 152) by about 4.96% and 1.99%, respectively. Experimental results show that compared with the original model, the new network improves the recall rate and achieves better detection performance.

In addition, the model in this paper is superior to such regression-based algorithms as the RRD (rotation-sensitive regression detector) algorithm in the evaluation index. Ours-ResNet-18 (1280 × 736) outperforms the EAST algorithm by about 3.66%, 5.98%, and 4.99% in accuracy, recall, and F value respectively. The Corner algorithm will predict two adjacent texts as one text instance, resulting in inaccurate detection [34]. SPN algorithm (Short Path Network) has poor robustness for curved text examples. When the candidate region predicted in the first stage only contains a part of the text instance, the SRPN (Scale-based Region Proposal Network) algorithm cannot correctly predict the boundary of the whole text instance in the second stage [35]. Compared with the EAST algorithm, Corner algorithm, SPN algorithm, and SRPN algorithm, our model makes full use of semantic information to improve the accuracy of text pixel prediction and classification, reduces the interference of background pixels on small-scale texts, and makes use of rich feature information to improve the positioning ability of text examples.

On the long-text dataset MSRA-TD500, the comparison results between our algorithm and other algorithms are shown in Table 8. Ours-ResNet-18 (512 × 512) has achieved 77.15% recall rate, 90.16% accuracy rate, and 83.15% F value, which exceeds the original model DB-ResNet-18 (512 × 512) by about 4.46% and 3.95%, respectively. Ours-ResNet-18 (736 × 736) achieved a recall rate of 83.33%, an accuracy rate of 88.02%, and an F value of 85.61%, which surpassed the original model DB-ResNet-18 (736 × 736) by about 7.63% and 2.81%, respectively. Ours-ResNet-50 (736 × 736) achieved 84.71% recall rate, 89.80% accuracy rate, and 87.18% F value, which exceeded the original model DB-ResNet-50 (736 × 736) by about 5.51% and 2.28% respectively, exceeding the table. In addition, Ours-ResNet-18 (736 × 736) outperforms the segmentation-based TextSnake algorithm and CRAFT algorithm in three evaluation indexes.

Figure 7 shows the visualization results of our method and the original DBNet on different types of text examples. It is worth noting that the images here are randomly selected from three datasets, which can better prove the robustness of our model.

For Figure 7a, comparing Baseline and Ours, Baseline missed a part of the text in the figure (i.e., “CA”), while our method can detect it. For Figure 7b,c, Baseline mistakenly detects the non-text area and detects the non-text area as the text area. Compared with Baseline, our method can avoid the false detection. As for Figure 7d, comparing Baseline and Ours, Baseline missed a part of the text (i.e., “1”) in the figure, while our method can detect it. For Figure 7e, Baseline missed the middle English text, but our method can accurately detect it. For Figure 7f, Baseline detects “COFFEE” as two parts of text, but the actual “COFFEE” represents the same semantic information, which should be detected as a whole text area, and our method can detect it.

The above results show that the proposed algorithm improves the detection ability on the multi-directional text dataset ICDAR2015, curved text dataset Total-Text, and multilingual text dataset MSRA-TD500. We can see that our network is very good in the natural scene text detection dataset, with good accuracy, recall rate, and F value, and can obtain a more efficient network. Experiments show that the residual correction branch (RCB) and double branch attention feature fusion (TB-AFF) modules are very important for text feature extraction and location information enhancement, which can improve the detection accuracy of the original algorithm without losing the detection efficiency. At the same time, in various challenging scenes, such as uneven lighting, low resolution, complex background, etc., this model can effectively deal with the drastic scale change of text, effectively improve the effect of text detection in natural scenes and accurately detect the scene text, which to some extent is inseparable from our proposed network model.

## 5. Discussion and Conclusions

In this chapter, based on ResNet and FPN network, a scene text detection algorithm based on double-branch feature extraction is proposed. The Methods Residual Correction Branch (RCB) and Double Branch Attention Feature Fusion (TB-AFF) are used to extract lightweight features. In order to make up for the deficiency of the feature-extraction ability and the receptive field of a lightweight network, embedding residual correction branch (RCB) into the backbone network to enhance its feature-extraction ability is helpful to locate the position of the text area more accurately, without including too many background parts, even at low network depth. In addition, this paper also proposes a dual-branch attention feature fusion (TB-AFF) module, which is used to enhance the feature expression of multi-scale scene texts. It can make the network model extract features more efficiently, pay more attention to the label-related targets, improve the accuracy of its detection, effectively improve the existing models, and demonstrate good universality.

However, it is worth noting that exploring the setting of the best architecture is beyond the scope of this paper. This paper only makes a preliminary study on how to improve convolutional neural networks. We encourage readers to further study and design more effective structures and provide a different perspective of network-architecture design for computer vision. In the future, we will further optimize the structure of segmentation network and study a better network model, to reduce the complexity of the model, shorten the training time, improve the overall performance of the algorithm, and improve the accuracy of the deep learning model.

## Figures and Tables

**Figure 1 sensors-22-06262-f001:**
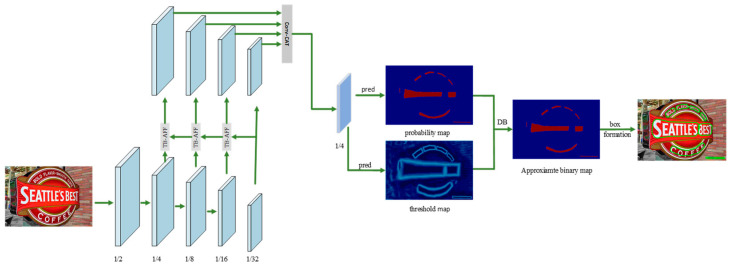
It shows the overall framework of the method proposed in this paper. In this paper, the backbone network ResNet is modified appropriately, and the residual correction branch is designed to improve the ability of network feature extraction. Secondly, a more efficient feature fusion module, namely, Two-Branch Attention Feature Fusion (TB-AFF) module, is adopted.

**Figure 2 sensors-22-06262-f002:**
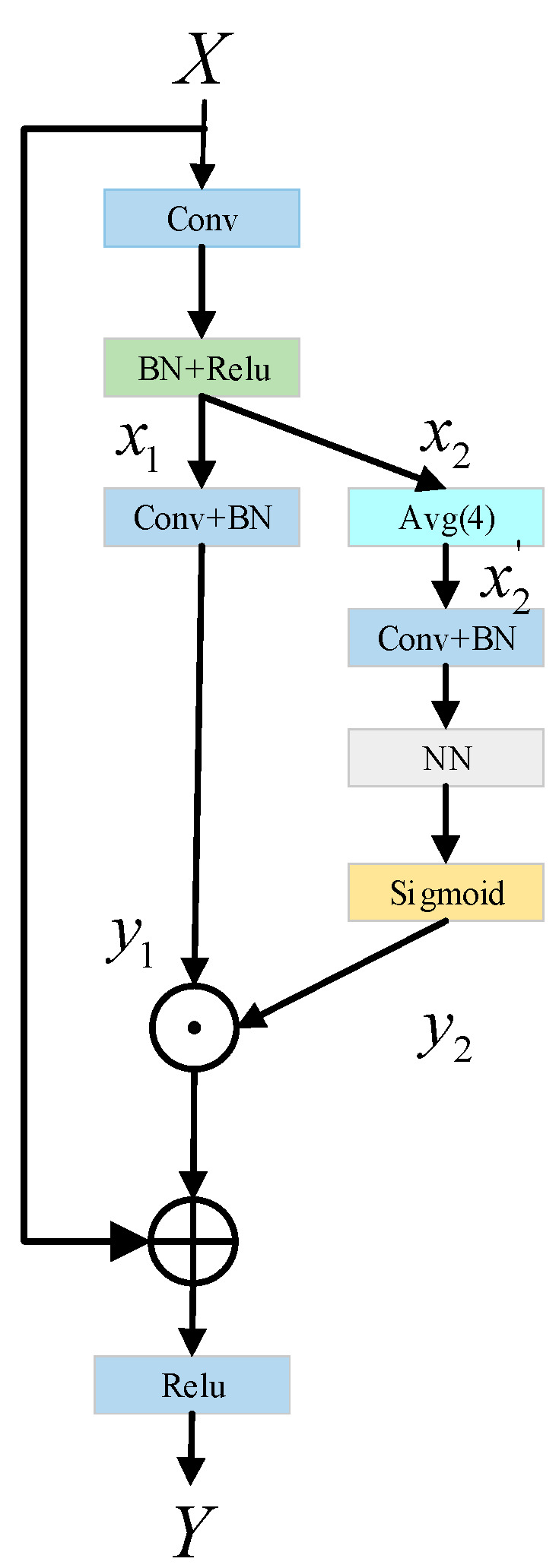
Structural design of residual correction branch (RCB).

**Figure 3 sensors-22-06262-f003:**
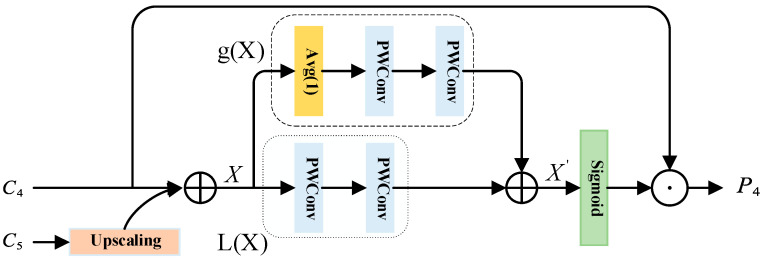
Structure design of two-branch attention feature fusion (TB-AFF) module.

**Figure 4 sensors-22-06262-f004:**
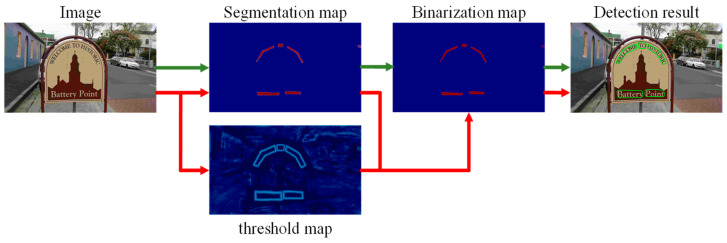
Structure diagram of differentiable binarization.

**Figure 5 sensors-22-06262-f005:**
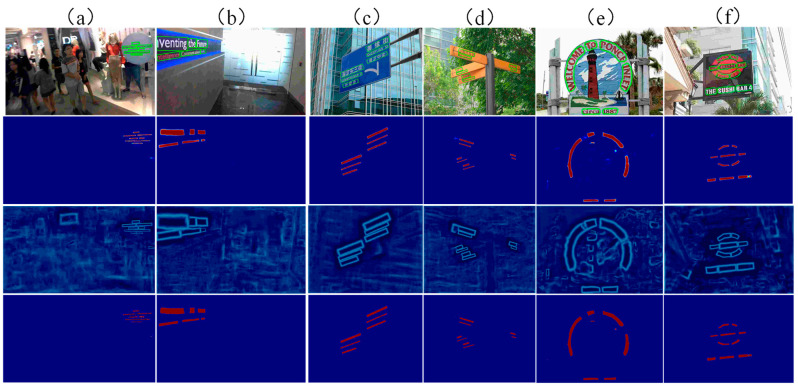
Visualization results of different types of text examples. Where (**a**,**b**) are multi-directional texts, (**c**,**d**) are multi-lingual texts, (**e**,**f**) are curved texts.

**Figure 6 sensors-22-06262-f006:**
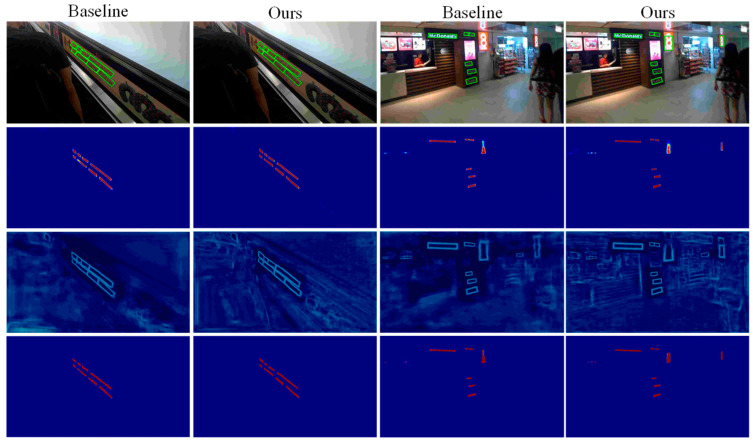
Visualization results of baseline and ours.

**Figure 7 sensors-22-06262-f007:**
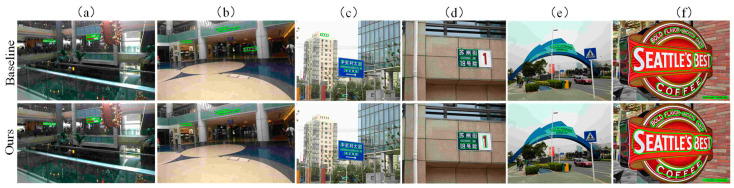
Visualization results display on different types of text examples. Where (**a**,**b**) are multi-directional texts, (**c**,**d**) are multi-lingual texts, (**e**,**f**) are curved texts.

**Table 1 sensors-22-06262-t001:** Size of feature maps.

	H×W×C		H×W×C
Y_2_	320×320×64	C_2_/P_2_	320×320×64
Y_3_	160×160×128	C_3_/P_3_	160×160×64
Y_4_	80×80×256	C_4_/P_4_	80×80×64
Y_5_	40×40×512	C_5_/P_5_	40×40×64

**Table 2 sensors-22-06262-t002:** Test results in ICDAR 2015 dataset.

Method	RCB	TB-AFF	P (%)	R (%)	F (%)
Resnet-18			89.3	73.80	80.80
Resnet-18	√		86.66	78.48	82.36
Resnet-18		√	87.51	78.62	82.83
Resnet-18	√	√	87.26	79.48	83.19

**Table 3 sensors-22-06262-t003:** Test results in Total-Text dataset.

Method	RCB	TB-AFF	P (%)	R (%)	F (%)
Resnet-18			86.70	75.4.	80.70
Resnet-18	√		88.05	78.36	82.92
Resnet-18		√	87.04	79.13	82.90
Resnet-18	√	√	87.37	78.95	82.95

**Table 4 sensors-22-06262-t004:** Test results in MSRA-TD 500 dataset.

Method	RCB	TB-AFF	P (%)	R (%)	F (%)
Resnet-18			85.70	73.20	79.0
Resnet-18	√		86.84	81.62	84.15
Resnet-18		√	87.17	80.58	83.75
Resnet-18	√	√	88.02	83.33	85.61

**Table 5 sensors-22-06262-t005:** Ablation experiment of sampling factor r.

Method	Flops(G)	P (%)	R (%)	F (%)
Ours-Resnet-18 (r = 1)	45.44	86.45	79.87	83.03
Ours-Resnet-18 (r = 2)	39.77	86.79	79.73	83.11
Ours-Resnet-18 (r = 3)	38.65	87.63	79.15	83.17
Ours-Resnet-18 (r = 4)	38.35	87.26	79.48	83.19
Ours-Resnet-18 (r = 5)	38.18	87.97	78.86	83.17

**Table 6 sensors-22-06262-t006:** Test results of curved text dataset. The value in brackets refers to the height of the input image. “*” means multi-scale test. “MTS” and “PSE” are the abbreviations for Mask TextSpotter and PSENet.

Method	Params (M)	Flops (G)	P (%)	R (%)	F (%)
TextSnake (Long et al., 2018)	19.1	136.01	82.7	74.5	78.4
ATRR (Wang et al., 2019b)	-	-	80.9	76.2	78.5
MTS (Lyu et al., 2018a)	-	-	82.5	75.6	78.6
TextField (Xu et al., 2019)	-	-	81.2	79.9	80.6
LOMO (Zhang et al., 2019) *	-	-	87.6	79.3	83.3
CRAFT (Baek et al., 2019)	20.8	146.29	87.6	79.9	83.6
CSE (Liu et al., 2019b)	-	-	81.4	79.1	80.2
PSE-1s (Wang et al., 2019a)	28.6	117.1	84.0	78.0	80.9
**DB-ResNet-18 (800 × 800)**	12.2	24.46	86.7	75.4	80.7
**Ours-ResNet-18 (800 × 800)**	18.4	38.35	87.37	78.95	82.95
**DB-ResNet-50 (800 × 800)**	25.5	49.36	84.3	78.4	81.3
**Ours-ResNet-50 (800 × 800)**	69.9	60.99	88.06	82.19	85.03

**Table 7 sensors-22-06262-t007:** Test results of ICDAR 2015 dataset (the values in brackets indicate the height of the input image).

Method	P (%)	R (%)	F (%)
EAST (Zhou et al., 2017)	83.6	73.5	78.2
Corner (Lyu et al., 2018b)	94.1	70.7	80.7
RRD (Liao et al., 2018)	85.6	79.0	82.2
PAN (Wang et al., 2019)	84.0	81.9	82.9
PSE-1s (Wang et al., 2019a)	86.9	84.5	85.7
SPCNet (Xie et al., 2019a)	88.7	85.8	87.2
LOMO (Zhang et al., 2019)	91.3	83.5	87.2
CRAFT (Baek et al., 2019)	89.8	84.3	86.9
SAE (Tian et al., 2019)	88.3	85.0	86.6
SRPN (He et al., 2020)	92.0	79.7	85.4
**DB-ResNet-18 (1280 × 736)**	89.3	73.8	80.8
**Ours-ResNet-18 (1280 × 736)**	87.26	79.48	83.19
**DB-ResNet-50 (1280 × 736)**	88.6	77.8	82.9
**Ours-ResNet-50 (1280 × 736)**	87.82	79.83	83.63
**DB-ResNet-50 (2048 × 1152)**	89.8	79.3	84.2
**Ours-ResNet-50 (2048 × 1152)**	88.21	84.26	86.19

**Table 8 sensors-22-06262-t008:** Test results of the algorithm on MSRA-TD500 dataset (the value in brackets is the height of the input image).

Method	P (%)	R (%)	F (%)
(He et al., 2016b)	71.0	61.0	69.0
EAST (Zhou et al., 2017)	87.28	67.43	76.08
DeepReg (He et al., 2017b)	77.0	70.0	74.0
SegLink (Shi et al., 2017)	86	70	77
RRPN (Ma et al., 2018)	82.0	68.0	74.0
RRD (Liao et al., 2018)	87.0	73.0	79.0
MCN (Liu et al., 2018)	88.0	79.0	83.0
PixelLink (Deng et al., 2018)	83.0	73.2	77.8
Corner (Lyu et al., 2018b)	87.6	76.2	81.5
TextSnake (Long et al., 2018)	83.2	73.9	78.3
(Xue, Lu, and Zhan 2018)	83.0	77.4	80.1
(Xue, Lu, and Zhang 2019)	87.4	76.7	81.7
CRAFT (Baek et al., 2019)	88.2	78.2	82.9
SAE (Tian et al., 2019)	84.2	81.7	82.9
PAN (Wang et al., 2019)	84.4	83.8	84.1
SRPN (He et al., 2020)	84.9	77.0	80.7
**DB-ResNet-18 (512 × 512)**	85.7	73.2	79.0
**Ours-ResNet-18 (512 × 512)**	90.16	77.15	83.15
**DB-ResNet-18 (736 × 736)**	90.4	76.3	82.8
**Ours-ResNet-18 (736 × 736)**	88.02	83.33	85.61
**DB-ResNet-50 (736 × 736)**	91.5	79.2	84.9
**Ours-ResNet-50 (736 × 736)**	89.80	84.71	87.18

## Data Availability

Total-Text dataset: https://github.com/cs-chan/Total-Text-Dataset (accessed on 10 January 2022). MSRA-TD500 dataset: MSRA Text Detection 500 database (MSRA-TD500)—TC11 (iapr-tc11.org (accessed on 10 January 2022)). ICADR2015 dataset: Tasks—Incidental Scene Text—Robust Reading Competition (uab.es (accessed on 10 January 2022)).

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
