# Peer review of "Scene Text Detection Based on Two-Branch Feature Extraction"

_sensors, 2022, doi:10.3390/s22166262_

Round 1

Reviewer 1 Report

The convolution neural network improvement solutions used by the authors, which were applied to scene text detection, significantly help the machine interpretability of texts of any direction and shape. 

Based on all this, the method presented by the authors can further promote the development of new methods that can lead to an even faster and more accurate scene text detection.

Compared to the ones used so far (EAST, DeepReg, DB-ResNet-18, etc.), the method they developed shows a serious improvement in terms of accuracy, recallability and comprehensive evaluation index when performing the 3 broad sample tasks they used.

I found only one problem in the manuscript, namely that I could not find figure 2, which would be extremely important for a more precise understanding, in the text. By replacing this, the manuscript is, in my opinion, suitable for publication.

Reviewer 2 Report

1. How big is the SynthText dataset? Why was the model pre-trained on it only for 100k iterations? Why 1200 iterations for fine-tuning? Did authors do any loss curve experiments on validation set to fix the iterations numbers?
2. Are other models in Table-4 also pre-trained on SynthText dataset? If not then it would be unfair to compare them with the current model which leverages extra information from another dataset.
3. Please include the Model size and training times for all models in Table 4 for a fair comparison of models?
4. What are the output dimensions at each level of feature pyramid network used for this model (C and P branches)?
5. In TB-AFF module, authors mention that X is generated by simple addition of C4 and C5. Are there same number of channels in C4 and C5 for them to be added element-wise?
6. In Figure 3 TB-AFF module, the feature map P4 is generated by using the fusion feature X and its averaged version. What are the effects of adding multiple branches in TB-AFF which compute different downsampled versions of X instead of just having the original and averaged X?
7. In Residual Connection Branch, authors downsample the original image using a 4x4 window (line 246-247). Why 4x4 window is used? Did authors try other kernel sizes? How about including information from multiple resolution/downsampling window sizes?
8. Did authors experiment with using simply SENet as attention module? Does TB-AFF have extra advantage over SENet in terms of perfromance improvement?

Author Response

This manuscript is a resubmission of an earlier submission. The following is a list of the peer review reports and author responses from that submission.